# The Role of Promyelocytic Leukemia Zinc Finger (PLZF) and Glial-Derived Neurotrophic Factor Family Receptor Alpha 1 (GFRα1) in the Cryopreservation of Spermatogonia Stem Cells

**DOI:** 10.3390/ijms24031945

**Published:** 2023-01-18

**Authors:** Asma’ ‘Afifah Shamhari, Nur Erysha Sabrina Jefferi, Zariyantey Abd Hamid, Siti Balkis Budin, Muhd Hanis Md Idris, Izatus Shima Taib

**Affiliations:** 1Center of Diagnostics, Therapeutics, and Investigative Studies (CODTIS), Faculty of Health Sciences, Universiti Kebangsaan Malaysia, Jalan Raja Muda Abdul Aziz, Kuala Lumpur 50300, Wilayah Persekutuan, Malaysia; 2Integrative Pharmacogenomics Institute (iPROMISE), Universiti Teknologi MARA (UiTM), Puncak Alam Campus, Bandar Puncak Alam 42300, Selangor, Malaysia

**Keywords:** antioxidant, cryopreservation, infertility, spermatogonia, spermatogenesis, GFRα1, PLZF

## Abstract

The cryopreservation of spermatogonia stem cells (SSCs) has been widely used as an alternative treatment for infertility. However, cryopreservation itself induces cryoinjury due to oxidative and osmotic stress, leading to reduction in the survival rate and functionality of SSCs. Glial-derived neurotrophic factor family receptor alpha 1 (GFRα1) and promyelocytic leukemia zinc finger (PLZF) are expressed during the self-renewal and differentiation of SSCs, making them key tools for identifying the functionality of SSCs. To the best of our knowledge, the involvement of GFRα1 and PLZF in determining the functionality of SSCs after cryopreservation with therapeutic intervention is limited. Therefore, the purpose of this review is to determine the role of GFRα1 and PLZF as biomarkers for evaluating the functionality of SSCs in cryopreservation with therapeutic intervention. Therapeutic intervention, such as the use of antioxidants, and enhancement in cryopreservation protocols, such as cell encapsulation, cryoprotectant agents (CPA), and equilibrium of time and temperature increase the expression of GFRα1 and PLZF, resulting in maintaining the functionality of SSCs. In conclusion, GFRα1 and PLZF have the potential as biomarkers in cryopreservation with therapeutic intervention of SSCs to ensure the functionality of the stem cells.

## 1. Introduction

Infertility has been described as a disease of the reproductive system by the World Health Organization (WHO) [1]. This disease is characterized by the inability to achieve clinical pregnancy after one year or more of regular sexual intercourse without any protection [2]. The two types of infertility that have been classified by most clinicians and researchers are primary infertility and secondary infertility. Primary infertility occurs when a couple is unable to conceive after at least 12 months of unprotected sexual activity, whereas secondary infertility refers to a couple’s inability to conceive after at least one previous successful pregnancy [2].

Infertility is equally common in men and women [3,4]. Infertility in men is due to several factors such as low sperm quality, problems with the delivery of sperms, overexposure to environmental agents, and damage related to cancer and its treatment [5,6,7]. One of the causes of low sperm quality is a defect in spermatogenesis. Spermatogenesis is defined as a process in which mature and functional sperm are synthesized from the spermatogonia stem cells (SSCs). Defects in spermatogenesis eventually lead to low sperm quality. Low sperm quality is characterized by a decrease in sperm count (oligospermia or azoospermia), decrease in sperm motility (asthenozoospermia), and deficiency in the structure/morphology of the sperm (teratozoospermia) [8,9]. However, people who are diagnosed with azoospermia experience a more traumatic treatment journey. Azoospermia is a condition where the semen does not contain any sperm, which is one of the causes due to spermatogenesis defects. Hence, the possibility of the patient being able to conceive a baby seems impossible. Furthermore, cancer treatments with gonadotoxicity can adversely affect fertility among cancer survivors. Exposure to irradiation and alkylating chemotherapeutic agents in cancer treatments for patients has been proven to reduce the reproductive health potential, resulting in the loss of SSCs [10,11].

It was predicted that 0.4% of young adults aged between 20 and 29 years will be long-term survivors of childhood cancer by 2010 [12,13]. After cancer therapy, young cancer survivors typically experience complications such as infertility. Hence, these infertility conditions can be treated via the restoration of sperms, involving the cryopreservation of SSCs. Preservation of the testis tissue before treatment offers a method for individuals whose spermatozoa cannot be retained because chemotherapy or radiation therapy commonly results in permanent infertility due to the loss of SSCs [13]. When spermatozoa cannot be maintained, cancer patients have the option to preserve their fertility by preserving their testis tissue prior to cancer therapy [14,15]. In order to minimize the necessity for assisted reproduction later in life, autologous transplantation of SSCs back to the testis may produce spermatozoa [14,16].

However, it is essential to know that long-term cryopreservation of testicular tissue may have negative effects on the viability and functionality of SSCs. Several studies have shown that the survival rate of SSCs and the quality of differentiated germ cells are lower in frozen-thawed tissue than in fresh controls, indicating that cryopreservation protocols must be improved [17,18,19]. Cryopreservation could also cause adverse cryo-injuries, which disrupt the normal biological activity of cells. In addition, past studies revealed that the functional capability of thawed SSCs is low after cryopreservation. The alterations include DNA breaks, mitochondrial malfunction, osmotic stress, oxidative stress [20], activation of apoptosis [21], and an increase in the production of reactive oxygen species (ROS) [22,23]. It is generally known that the cryopreservation technique promotes the generation of ROS and oxidative stress, which can damage the biological components such as membranes, structural proteins, enzymes, and nucleic acids [22,23]. Cryopreservation was also found to caused damage to key tissues and cellular activities, thus increasing the risk of cell death [24,25].

The population of SSCs may undergo several changes in terms of their behavior (morphological aspects and protein expression) during spermatogonia regeneration and stem cells potential (self-renewal and differentiation). During the cryopreservation of SSCs, the molecular proteins such as glial cell line-derived neurotrophic factor receptor alpha 1 (GFRα1) and promyelocytic leukemia zinc finger (PLZF) are the crucial markers for the functionality of SSCs [26]. The levels of GFRα1 and PLZF were found to be significantly increased during spermatogenesis wherein both markers are crucial for self-renewal [26,27,28,29,30,31]; however, only PLZF is involved in initiating the differentiation of spermatogonia [32,33,34,35,36]. Cryopreservation itself causes cryoinjury, resulting in the reduction in the survival rate and functionality of SSCs. Cryoinjury could occur due to oxidative and osmotic stress, and these events are caused by the increase in the physiological production of ROS and owing to the limitations of the cryopreservation protocol, respectively. This occurrence prompted researchers to examine and conduct more experiments to enhance the survival rate and functionality of SSCs by adding supplementations such as antioxidants to the media and modifying the cryopreservation techniques. GFRα1 and PLZF are among the molecular proteins that are expressed during SSC self-renewal and differentiation, which can be considered a significant tool for the isolation of potential SSCs. The cryopreservation of SSCs can be utilized as a novel restorative material for individuals who wish to treat and preserve their fertility status via cryopreservation. To the best of our knowledge, the role of GFRα1 and PLZF for assessing the functionality of SSCs in cryopreservation with therapeutic intervention is limited. Therefore, this review was performed to explore the role of GFRα1 and PLZF as biomarkers for assessing the SSC functionality in cryopreservation with therapeutic intervention to optimize the efficiency of cryopreservation techniques.

## 2. Spermatogenesis

Spermatogenesis is required for the development of mature sperm cells. It is a complicated, time-based event in which primitive and totipotent stem cells divide to either renew themselves or differentiate into daughter cells. These events will lead to the formation of testicular spermatozoa. Spermatogenesis is regulated by the hypothalamus–pituitary–gonad (HPG) axis. The hypothalamus secretes the gonadotropin-releasing hormone (GnRH) that stimulates the anterior pituitary gland to release the follicle-stimulating hormone (FSH) and luteinizing hormone (LH) [37,38]. LH and FSH circulate in the body via blood and trigger specific cell types in the testes, which are the Leydig cells and Sertoli cells, respectively. Leydig cells produce testosterone, which is the main androgen hormone that regulates spermatogenesis in the testis. In addition, testosterone is required for the completion of meiosis during spermatocyte development and serves as a crucial functional molecules in delaying the release of the elongated spermatozoa. Furthermore, this hormone is also essential for the SSCs to be functional after cryopreservation [39]. Meanwhile, FSH modulates Sertoli cell function via the FSH receptor (FSHR) to stimulate spermatogenesis. FSH is also believed to function independently and in conjunction with testosterone to increase the Sertoli cell proliferation and create signaling molecules and nutrients to enhance spermatid maturation [40]. In the post pubertal testis, FSH and testosterone trigger Sertoli cells to promote germ cell maturation, supply anti-apoptotic survival factors, and control adhesion complexes between germ cells and Sertoli cells [41,42]. Therefore, LH and FSH signaling is essential for initiating and sustaining spermatogenesis. Specifically, spermatogonia, primary spermatocytes, secondary spermatocytes, and spermatids are arranged in a highly ordered sequence from the basement membrane to the lumen of the seminiferous tubule [43,44]. These germ cell types are associated with spermatogenesis events such as proliferation, differentiation, meiosis, and spermiogenesis, a complex process that transforms round spermatids into complex spermatozoa following meiosis. In mice, spermatogenic differentiation is driven by a series of mitotic divisions that originate from type A_single_ spermatogonia and is later derived from A_paired_, A_aligned_, A_1_, A_2_, A_3_, and A_4_ spermatogonia [45,46]. Type A_single_ spermatogonia are the self-renewing SSCs that give rise to type A_paired_ and A_aligned_ spermatogonia. Continued mitotic expansions produce A_1_–A_4_ spermatogonia, followed by the formation of type B and intermediate spermatogonia, resulting in the formation of huge and linked cohorts of spermatogonia [45,47]. As initially postulated by Clermont [48], in primates including humans, self-renewing and differentiated SSCs are represented by type A_dark_ and type A_pale_ spermatogonia, respectively. Several studies also reveal the presence of a transitioning spermatogonial population (25–50% of the spermatogonial population) that is morphologically transitional and distinct from type A_dark_ and A_pale,_ spermatogonia and these are referred to as type A_transition_ spermatogonia [35]. Type A_pale_ spermatogonia then proceeds to develop into type B spermatogonia. Type B spermatogonia will undergo cell division to differentiate into primary spermatocytes, secondary spermatocytes, and spermatids. From the spermatids, spermiogenesis occurs where spermatozoa are released into the lumen of the seminiferous tubules. Following puberty, once the baseline number of spermatogonia has been established, the mitotic component continues to produce precursor cells and initiates the process of differentiation and maturation of the germ cells [49,50].

The ability to self-renew is a distinctive characteristic of type A_single_ and type A_dark_ spermatogonia. However, a theory for stem cells has been developed in which a heterogeneous pool of SSCs (type A_single_ spermatogonia or type A_dark_ spermatogonia, type A_paired_ spermatogonia, or type A_pale_ spermatogonia) permits individual cells to respond differently based on their current marker profile. Induction by microenvironmental signals and the actual condition of the individual cell results in multiple cell fate options, including self-renewal, death, and differentiation [51,52]. This new stem cell model suggests that cells do not follow a unidirectional differentiation process. Applying this theory to germ cells, spermatogonia may not develop unidirectionally and sequentially from stem cells to differentiated B spermatogonia, but rather may switch between many spermatogonia subtypes [45,52].

The SSCs are characterized and classified based on their clonal size, clonal arrangement, molecular markers, and their biological function after the transplantation procedure [53]. However, this purely morphological characterization and classification were challenged by the application of molecular, histological, and functional markers [54]. Currently, fluorescence-activated cell sorting (FACS) has been used to further characterize colonizing primate stem cell subpopulations based on the characterization of spermatogonia in rodents [45]. Based on molecular marker profiles by using the FACS method, different subpopulations of primate spermatogonia representing type A_dark_ spermatogonia were identified, where it had the most undifferentiated phenotypic profile, i.e., [glial cell-line derived neurotrophic factor (GDNF) family receptor alpha-1+/promyelocytic leukemia zinc finger+/KIT proto-oncogene receptor tyrosine kinase-(GFRα1+/PLZF+/cKIT−)]. The expression of PLZF is constant in the type A_single_, type A_paired_, and type A_aligned_ spermatogonia (in mice) or type A_dark_ and type A_pale_ spermatogonia (in primates and humans) and has been used to identify all undifferentiated spermatogonia, while GFRα1 is shown to be preferentially expressed in type A_single_ spermatogonia (type A_dark_ spermatogonia) [45,55,56].

GFRα1 is a self-renewal marker without differentiation potential, while PLZF is a self-renewal marker with differentiating potential [9,57,58]. It has been proven by the previous studies that the absence and mutation of PLZF markers are found to cause gradual loss of germ cells [27], testicular hypoplasia [33], and infertility [31]. A disruption in the function of the SSC population results in the failure of spermatogenesis. Additionally, when the syncytial length increases, the expression of GFRα1 within type A_undiff_ spermatogonia decreases. While roughly 90% of type A_single_ spermatogonia express GFRα1, nearly 75% of type A_paired_ spermatogonia and approximately 40–15% of type A_1_ spermatogonia express GFRα1 [56,59,60]. Similar findings were found where mutations in *GFRα1* in mammals caused inhibition of SSCs proliferation, resulted in decreasing cell populations [35]. Hence, these markers are important to monitor the regulation of spermatogenesis. Therefore, PLZF and GFRα1 are essential to evaluate the function and integrity of SSCs.

## 3. Spermatogonia Stem Cell Biomarkers

Different phenotype subpopulations of SSCs might be identified in terms of their differentiation stages based on the molecular biomarker profiles [45,61]. Previous studies showed that the SSCs were found to be heterogenous after evaluating the biomarkers by using genetic labeling, lineage tracing analysis, and live imaging [60,61]. This indicated that the population of SSCs may undergo a change in terms of their behavior during regeneration and considering stem cell potential. Thus, the molecular biomarker can be considered a significant tool for isolating potential SSCs and developing the cell culture method from animal to human studies. The number of specific biomarkers for SSCs are reported to increase. GFRα1 and PLZF are found to be significant markers during spermatogenesis, where both markers are crucial for self-renewal [27,28,29,30,31], while only PLZF is involved in initiating the differentiation of spermatogonia [33,34,35,36,62].

### 3.1. GFRα1

Glial-derived neurotrophic factor family receptor alpha 1 (GFRα1) is a ligand-binding domain of the glial cell line-derived neurotrophic factor (GDNF) receptor that is substantially expressed in human A_pale_ and A_dark_ spermatogonia [63]. GFRα1 is a co-receptor of rearranged during transfection (RET) for the GDNF [64]. GDNF is essential for the neuron development of peripheral and central neurons [65] as well as kidney morphogenesis [66]. Additionally, GDNF regulates the proliferation and differentiation of SSCs [67,68]. For GDNF to elicit their physiological response, they require the presence of glycosyl phosphadidoinositol (GPI)-linked protein GFRα1. This complex will selectively bind to GDNF and promote the activation of the RET receptor to trigger intracellular signal cascades for the cell’s proliferation or self-renewal [64,69]. GDNF alone reduced the SSC number derived from C57 ROSA mice over a 10-week culture period, whereas GDNF with the addition of soluble GFRα1 enhanced the number of the cells, suggesting that GFRα1 plays a role in the GDNF-induced proliferation or self-renewal of SSCs [70].

When GDNF binds to its receptor complex, two primary signaling pathways are activated in undifferentiated spermatogonia. In one of the mechanisms, RET phosphorylation stimulates the activation of PI3K/AKT and SRC-family kinases (SFKs), allowing spermatogonia proliferation [71,72]. Recent studies revealed that the mTORC1 pathway is activated downstream of AKT as spermatogonia differentiate from A_aligned_ to A_diff_ [73], and that only AKT3 is phosphorylated in response to GDNF in undifferentiated spermatogonia [74]. The classical RAS/ERK1/2 (MAPK) pathway is the other pathway that is activated by the binding of GDNF to the RET/GFRA1 complex [75,76]. Nevertheless, it is still unclear which of these mechanisms, or even both, directly cause self-renewal or differentiation. Previous studies have demonstrated that RET tyrosine kinase inactivation occurs due to downregulation of Gfra1, which may subsequently disrupt the intracellular GDNF/GFRA1/RET signaling pathway [64,77]. In fact, this molecular mechanism causes the proliferation of SSCs to be inhibited and allows them to participate in the differentiation process. Therefore, the researchers anticipated that the elimination or suppression of GFRα1 expression might induce SSCs differentiation [78,79]. Hence, it has been proven that GFRα1 is crucial for self-renewal of SSCs but not the differentiation.

It has been proposed that GFRα1 and RET receptors as well as GDNF are essential for spermatogenesis [80]. Interestingly, these proteins are also involved during the development of the male embryonic gonad. It is proven that Gdnf, Ret, and Gfrα1 messenger ribonucleic acid (mRNA) expression is high during the development of the male embryonic gonad as early as embryonic day 12.5 (E12.5) in mice. However, GDNF was found to be reduced starting at E14.5, accompanied with apoptosis of the germ cells in Ret knockout embryos [81]. Another type of GDNF family ligand (GFL), persephin (PSPN), which is extensively expressed between E12.5 and E15.5, could compensate for the reduced GDNF protein [81]. Hence, the function of GDNF from the time of mitotic arrest for prospermatogonia at E15.5 until birth may be limited to germ cell survival. Prospermatogonia then migrates to the basement membrane soon after birth and develops into established SSCs. Then, GDNF binds to the RET/GFRα1 receptor complex and triggers the cells’ ability to self-renew forming A_paired_ and some A_aligned_ spermatogonia [82]. GFRα1 and RET are concurrently expressed in A_paired_ and some A_aligned_ spermatogonia [74]. Therefore, it is likely that GDNF will also stimulate and maintain the proliferation of these A type of spermatogonia cells before SSCs begin to differentiate [80].

The type A_undiff_ spermatogonia containing GFRα1 plays two important roles: 1) retain the SSC population; 2) transform into undifferentiated spermatogonia cells that contain NGN3^+^. In order to transform to undifferentiated spermatogonia containing NGN3^+^, Wnt signaling mediated by b-catenin is necessary for this mechanism [83]. The undifferentiated spermatogonia containing NGN3^+^ eventually expressed the retinoic acid (RA) receptor gamma (RARγ) for the development of differentiated spermatogonia (KIT+) cells in response to the RA pulse, which occurs once every 8.6-day cycle of seminiferous epithelium. These cells then undergo a series of mitotic divisions before meiosis [84,85,86]. However, the undifferentiated spermatogonia containing NGN3^+^ have the ability to transform back into undifferentiated spermatogonia that consists of GFRa1+ for self-renewal, which becomes prominent in regeneration following injury or transplantation [60,87].

Moreover, several studies have been performed to strengthen the theory of GFRα1 as a self-renewal marker in SSCs [28,88]. Mao et al. [88] revealed that the expression of GFRα1 was found to be higher in a subpopulation of SSCs (type A_dark_ or type A_single_ spermatogonia), which is located along the basement membrane in *Macaca fascicularis*. The SSCs from *Macaca fascicularis* were cultured and transplanted into the W mutant mice (kit mutation). After one month of transplantation, the GFRα1 expression was significantly detected via immunohistochemistry (IHC) along the basement membrane of the seminiferous tubules [88]. This indicated that GFRα1 has an important role in the proliferation and self-renewal of SSCs in maintaining spermatogenesis. A similar finding was also reported in the basement membrane of the seminiferous tubules in adult mice, wherein GFRα1 was found to be scattered unevenly in the whole-mount seminiferous tubule specimens using immunofluorescence staining [28]. The researchers found that GFRα1 was expressed in various SSCs such as type A_single_, A_paired_, and A_aligned-4_ spermatogonia; however, the highest GFRα1 expression was recorded in type A_single_ spermatogonia, which is crucial for self-renewal.

In addition, findings regarding the functional role of GFRα1 in SSCs of non-mammalian vertebrates, such as fish, remain scarce. However, a notable list of several studies revealed that fish species might share similar GDNF/GFRα1 signaling pathways with mammals [89,90]. Two Gdnf homologs, designated Gdnfa and Gdnfb, exist in medaka (*Oryzias latipes*), and both can mediate the self-renewal of SG3, an SSC line generated from adult medaka testis [89,90]. The medaka *gfra1* duplicate genes, *Olgfra1a* and *Olgfra1b*, were found to exhibit sequence and chromosome synteny as well as expression and function which are homologous to mammalian SSCs. By binding with Gdnfa and/or Gdnfb, respectively, OlGfra1a and OlGfra11b are both necessary for the self-renewal and maintenance of SSCs. These results corroborated that OlGfr1a and OlGfr11b may act similarly to their mammalian counterparts in SSCs. Therefore, this can be concluded that the expression of GFRα1 could be the self-renewal marker for evaluating the function of SSCs, particularly for the self-renewal capability in both mammalian and non-mammalian species.

### 3.2. PLZF

Promyelocytic leukemia zinc finger (PLZF) also known as ZBTB16 or ZFP145 was initially found in promyelocytic leukemia as one of the numerous partner proteins attached to the retinoic acid receptor α (RARα) by a reciprocal chromosomal translocation [91,92]. PLZF is a transcription factor that belongs to the poxviruses and zinc-finger (POZ)-Krüppel (POK) family; PZLF binds to specific DNA sequences via its carboxy-terminal zinc fingers. This specific binding towards DNA sequences could inhibit transcription by recruiting co-repressors via its amino terminal POZ domain. However, PLZF is also able to stimulate transcription depending on the activator and suppressor [35,93]. According to previous studies, PLZF was found to be involved in diverse signaling as well as differentiation and growth-regulatory pathways [58]. Despite PLZF being involved in the immune response system, this biomarker is the main target regulator in myeloid development [94,95]. Moreover, PLZF plays a major role in preserving and self-renewal in the population of stem cells [96].

In addition, PLZF is found to be highly expressed in type A_undiff_ spermatogonia at an early differentiation stage [33,62,77]. Thus, PLZF is a key cell-autonomous factor in promoting the ability of SSCs to self-renew [33] and a marker for differentiating subpopulation of spermatogonia. In mouse SSCs, PLZF works in at least three distinct ways to maintain SSCs, as follows: (i) by modulating the activity of SALL4 and c-kit, whose action is associated with spermatogonia differentiation [77,97,98], (ii) by directly and indirectly (via forkhead box protein O1 (Foxo1) and Etv5) repressing differentiation genes (including c-Kit) and stimulating spermatogonia stemness genes for self-renewal (GDNF) [59], and (iii) via indirectly inhibiting the differentiation ability via the mTORC1 pathway (Ddit4 over expression) [32,33]. During spermatogonia differentiation, there are interactions between PLZF and SALL4 proteins that share specific binding sites. The co-binding of PLZF and SALL4 was found to be correlated with active and passive regulatory mechanisms, leading to self-renewal and differentiation programs in undifferentiated spermatogonia [98]. Additionally, PLZF also binds to the promoter regions of Kit, Stra8, Sohlh2, and Dmrt1 to control the differentiation of stem and progenitor cells (SPCs), indicating a novel PLZF-mediated method for controlling SPC differentiation [99].

Through the regulation of the PI3K–Akt axis, a centrosomal protein of 55 kDa (CEP55) controls a molecular switch between SSC self-renewal and differentiation. According to Sinha et al. [100], Foxo1 was rendered inactive by the overexpression of Cep55, which prevented spermatogonia from differentiating. Foxo1 regulates a network of putative downstream targets, including Kit and Ret, and is therefore necessary for both SSC self-renewal and differentiation. Foxo1 functionality is inactivated and degraded in germ cells as a result of Akt-dependent phosphorylation, which is connected to the transcription factor’s translocation from the nucleus to the cytoplasm [35]. Additionally, relative enrichment in Plzf^+^ spermatogonia cells was seen in association with Ret and Gfra1, along with concurrent down-regulation of early growth response-4 (Egr4). This revealed how the functional role of PLZF influences the entire population of germ cells via the modulation of CEP55 and Foxo1 in the preservation of spermatogonia homeostasis.

Based on a previous study, the self-renewal of SSCs could be characterized with the expression of PLZF associated with GDNF and the inhibition of the mTORCH1 pathway. GDNF belongs to the transforming growth factor-β (TGF-β) superfamily where it is highly expressed in the testes, kidneys, and stomach. In the testis, Sertoli cells and peritubular myoid cells are known to produce GDNF. The mTORCH1 pathway stimulation is important in the differentiation of SSCs. PLZF is able to inhibit the mTORCH1 pathway, thus eliminating the differentiation of SSCs and maintaining the self-renewal ability [33]. In the same study, the PLZF knockout mice displayed increasing expression of mTORCH1 and downregulation of GDNF [33]. Therefore, there is an association between the PLZF, mTORCH1, and GDNF signaling in regulating the self-renewal of SSCs.

Moreover, the role of PLZF in spermatogenesis was found to be linked with the proteasome system. In normal physiology, REG, a proteasome activator, is indeed necessary for the optimal progression of spermatogenesis via inhibition of p53, which negatively controls PLZF transcription. Gao et al. [101] performed a molecular investigation and showed that the absence of REGγ substantially increased the amount of p53 protein in the testes of REGγ knockout mice and directly inhibited PLZF transcription in cell lines, resulting in defective male subfertility [101]. PLZF is involved in two important functions: maintaining the self-renewal capability and stimulating the differentiation of SSCs, depending on the specific regulator and suppressor protein or signaling. Hence, the loss of functional PLZF results in progressive germ cell loss, testicular hypoplasia, and infertility [36,62,101].

Additionally, Sertoli cells and androgen levels are involved in the differentiation of SSCs. Wang et al. [102] found an increase in the differentiation of the SSCs during androgen stimulation. Androgen hormone binds to the androgen receptor and induces the signal transduction to the SSCs nucleus to turn off the PLZF via plzf siRNA. Turning off the PLZF initiates the differentiation process and decreases the population of undifferentiated SSCs. In a similar study, 6-week-old androgen-deprived male mice were injected with bicalutamide intraperitoneally [102]. The researchers found out that there was accumulation of undifferentiated spermatogonia, including the PLZF+ population, caused by androgen derivatives where it blocks the spermatogenesis at the stage where ckit+1-integrin+ spermatogonia develop into more differentiated populations. These findings suggest that blocking of the androgen will prevent the spermatogonia from differentiating into spermatocytes in adult mouse testes while simultaneously promoting the accumulation of SSCs (which stimulates self-renewal) [103]. Therefore, the PLZF plays a role in self-renewal and differentiation of SSCs; however, it is dependent on the specific signaling pathway. It is proven that the PLZF could be the best marker to measure the functionality and integrity of SSCs. Due to the uniqueness of SSCs, scientists utilize them as a novel restorative material for individuals who wish to treat and preserve their fertility status via cryopreservation.

## 4. Cryopreservation Experiments Involving Human and Animal SSCs

Currently, no optimal standard method has been established for the cryopreservation of SSCs for male fertility. The essential steps of a cryopreservation procedure can be summed up as follows: (i) tissue preparation; (ii) addition of cryoprotectant agents to cells or tissues prior to freezing; (iii) freezing of the cells or tissues to a low temperature of −196 °C in liquid nitrogen; (iv) thawing of the cells or tissues; and (v) removal of the CPAs from the cells/tissues after thawing [104]. After that, cells or tissues can be clinically used for spermatogonial stem cells transplantation (SSCT), testicular tissue grafting, or in vitro spermatogenesis. Every step in the cryopreservation protocol has a significant impact on the success rate of restoring fertility, especially during the freezing and thawing procedures. However, during the tissue preparation step, the size of the testicular tissues does not significantly contribute to the success rate of restoring fertility. This is because, according to a previous study, the size of the original tissue fragment (5, 15, 20, or 30 mg) had no effect on the cell survival of immature pig testicular tissues following the same cryopreservation procedure [105]. Additionally, various fragment sizes (2–9 mm^3^) were effectively frozen for the cryopreservation of tissue from prepubertal boys [106,107,108,109,110,111]. However, Travers et al. [112] reported different findings in immature rat tissue where fewer morphological changes were observed in 7.5 mg compared to 15 mg of testicular tissue fragments by using similar cryopreservation procedures [112]. Furthermore, the success of cryopreservation also depends on the freezing and thawing steps. During the freezing step, the cellular metabolism halts, leading to elevated production of intracellular reactive oxygen species, organelle damage, and caspase-mediated apoptosis. All these events could lower the potential of sperm fertility [113]. This has been proven by previous studies that showed significant decreases in the morphology [114,115], motility [116], and DNA integrity [117,118] of post-thawed sperm. Developing the most efficient cryopreservation procedures focusing on freezing and thawing is crucial to enhance effective sperm retrieval and function.

Even though the success rate of fertility restoration after testicular tissue cryopreservation has yet to be demonstrated in humans, several centers around the world have collected and cryopreserved biopsied testicular tissues for future use [106,111,119,120,121,122]. Braye et al. [119] and Valli-Pulaski et al. [120] have performed long-term storage of cryopreserved testicular tissue from prepubertal and adult men. For the past 16 years (2002–2018), the Universitair Ziekenhuis in Brussels has frozen the testicular tissues of 112 patients between the ages of eight months and eighteen years. The testicular tissue was isolated from the patients in preparation for gonadotoxic cancer treatment (35%), gonadotoxic conditioning therapy for bone marrow transplantation (35%), or from boys with Klinefelter syndrome (KS) (30%). Valli-Pulaski et al. [120] had cryopreserved the testicular tissue from 189 patients between January 2011 and November 2018, for approximately eight years. The authors collected testicular tissue from patients ranging in age from 5 months to 34 years with a history of malignancies (*n* = 118), blood disorders (*n* = 45), and other disorders. Currently, there is an ongoing retrospective study by Sadri-Ardekani et al. [123] regarding the long-term storage of human SSCs by cryopreservation. The study collected immature testicular tissues from a prepubertal patient with cancer. However, until now, there is no evidence regarding the success rate of restoring germ cells after SSCs cryopreservation in any of the above-mentioned studies. Therefore, more extensive and comprehensive research studies are needed to confirm the safety and efficacy of SSCs cryopreservation, especially if it would be beneficial to treat and restore fertility.

Several studies have revealed the potential of testicular tissue cryopreservation in restoring fertility. A comparison between the use of fresh and cryopreserved testicular tissues indicated the positive findings for future uses. Pendergraft et al. [124] investigated the functionality of human SSCs after cryopreservation for 7 days, followed by culturing in media for 23 days. The findings showed that 52% of undifferentiated spermatogonia were able to express PLZF. de Michele et al. [125] collected immature testicular tissue from prepubertal patients and cryopreserved it for 21 days, followed by culturing it in media for 139 days. They revealed that the expression of GDNF, which is crucial in co-signaling with GFRα1 (biomarker for self-renewal of SSCs), does not significantly change as compared to that observed in fresh testicular tissue. These studies show the potential of human SSCs that can be cryopreserved. Not only that, the potential of testicular tissue cryopreservation has also been extensively studied in the context of testicular grafting, using both fresh [126,127,128] and cryopreserved [129,130,131] tissues in animal models such as pigs, mice, and monkeys. Before testicular tissue grafting was performed, spermatogenic development inside the grafted testicular tissue fragment was similar between fresh and frozen-thawed testicular tissues [132]. However, subsequent investigation found that the grafted frozen-thawed tissues had fewer intact tubuli at day one of grafting compared to the grafted fresh tissues, but at two months of grafting, there were no differences between these two types of transplanted tissues [133]. Furthermore, another study found that the frequency of Sertoli cell-only (SCO) tubules was higher in frozen-thawed testicular tissues than in fresh-thawed testicular tissue [134]. Besides, no differences were found between fresh and frozen-thawed testicular tissues in terms of the graft weight, stage of spermatogenesis, or percentage of tubuli with full spermatogenesis, regardless of the location of xenografting either ectopically or orthotopically [135]. Therefore, the previous research studies regarding the status of functionality of SSCs after the freeze-thawing procedures on human and non-human models proved that there is massive potential of testicular tissue cryopreservation for the future application of restoring fertility issues among men.

## 5. Clinical Application of SSCs Cryopreservation

Due to the gonadotoxic effects of cancer treatment, child cancer survivors are prone to have reproductive issues. Most chemotherapeutic drugs employ mechanisms that target proliferative cells, such as SSCs, causing severe damage to spermatogenesis. As high number of childhood cancer survivors risk infertility due to cancer treatment, these SSC-based therapeutic approaches may offer a considerable chance to recover the fertility and conceive biological children. To provide possibilities for fertility preservation and restoration, numerous approaches have been explored by utilizing the SSCs present in testicular tissues. However, for adult or postpubertal men, cryopreservation of ejaculated semen is the established and preferred way of conserving fertility, which is possible in around 90% of patients seeking fertility preservation. The semen was collected through masturbation techniques [136]. Cryopreservation of testicular tissue obtained via testicular biopsy or orchiectomy is now the only option for prepubertal patients in whom spermatogenesis is not yet initiated. Therefore, the SSC-based therapeutic approach is more well-known for prepubertal children than for adult men.

Owing to their ability to develop into sperm in vivo, SSCs are highly suitable for their potential in restoring fertility. Testicular tissue biopsies are cryopreserved for future clinical application of SSC-based therapeutic approaches until the patient desires to have children. After the cryopreservation of a testicular biopsy, several clinical therapeutic applications of SSC-based procedures can be attempted, such as SSCT, testicular tissue grafting, or in vitro spermatogenesis, either through cell-based or tissue-based culture. First, with SSCT, natural conception could be achieved [137]. Brinster et al. were the first to demonstrate that SSCT was effective in establishing colonization and spermatogenesis in the recipient testes of mice [138,139], and in producing offspring with the donor haplotype [138]. Since then, auto- or allo-transplantation of SSCs has been accomplished in a variety of mammals, including rodent and non-rodent species, as reported by Takashima and Shinohara [140], and in non-human primates [141,142], resulting in functional spermatogenesis in the recipient animals. SSCT clinical trials in humans have not yet been established; there is only one report of SSCT in seven men who received injections of cryopreserved testicular cells, but there is no subsequent report on the results of these therapies [143]. However, studies in mice have demonstrated that the number of transplanted SSC colonies decreases gradually during the homing process after transplantation [144]. Not only that, the number of transplanted SSCs colonies is highly dependent on the successful rate of SSCs colonization and donor-derived spermatogenesis within the recipient testis [144].

Second, the clinical application of SSC-based therapeutics approach also includes testicular tissue grafting. It is a procedure that involves the autologous grafting of immature testicular tissue fragments beneath the patient’s skin. The tissue gets endocrine signals from the body’s circulation through angiogenesis, hence facilitating spermatogenesis within the tissue [145,146]. After the completion of spermatogenesis, spermatids can be extracted from the recovered graft and utilized to fertilize eggs using assisted reproductive procedures such as intracytoplasmic sperm injection (ICSI). Testicular (xeno) grafting has been proven to result in effective spermatogenesis in testicular tissues of various mammalian species, including mice, pigs, and monkeys [127,128,147], resulting in the birth of live offspring after ICSI [126,128,135,148,149]. Recently, Fayomi et al. were successful in fertilizing primate oocytes with autologous primate graft spermatozoa, leading to the birth of a healthy rhesus macaque [135]. To date, no autologous grafting of human tissue has been reported; only xenografts in mice have been explored, although the use of human tissue in mouse xenografts has not yet resulted in a complete cycle of spermatogenesis [150,151].

Third, the clinical application of the SSC-based therapeutics approach includes in vitro spermatogenesis [152,153,154,155], wherein this procedure would have major clinical significance. However, the most difficult spermatogenic process to mimic in vitro is meiosis, the process by which two successive meiotic cell divisions form genetically distinct haploid spermatids. Although several in vitro studies documented the production of round spermatid-like cells from SSCs [152,153,155,156], only a few studies investigated the critical meiotic steps that are necessary for effective meiosis and mimicking real-derived gametes [153,155]. The lack of an appropriate testicular microenvironment, which is necessary to adequately sustain the spermatogenic process, may be the cause of the difficulties in mimicking spermatogenesis in vitro. Spermatogenesis requires spatial and temporal interactions between germ cells and somatic testicular cells in vivo. Incorporating a somatic niche in vitro could therefore be essential in sustaining the germ cells. Several crucial regulators, such as RA and gonadotropins, have been found in mice and humans in replicating the microenvironment that is necessary for spermatogenesis in vivo [157]. Hence, the three strategies such as SSCT, tissue grafting and in vitro spermatogenesis (Figure 1) for restoring fertility are still worthy options for researchers to explore further, ensuring that infertility is resolved in future generations.

## 6. Therapeutic Approach Improving the SSCs Cryopreservation

Cryopreservation of SSCs elicits similar cryoinjury damage as in sperm cryopreservation. Despite the damage caused by the process of freezing and thawing itself, the cryoinjury during cryopreservation is also associated with increase in ROS production due to the physiological event. The techniques used in cryopreservation, including drastic changes in temperature, ice crystal formation, and osmotic pressure along the cells, reduced the quality of SSCs, which may lead to unsuccessful preservation [158]. Extreme low and high cooling rates during cryopreservation could minimize the ability of the SSCs to regulate the intracellular water loss moderately [104]. Furthermore, this cryopreservation technique is also associated with the physiological ROS production where it could lead to a reduction in mitochondrial function [159]. Alterations in mitochondrial membrane fluidity increases the mitochondrial membrane potential, resulting in increased ROS production. The physiological production of ROS during cryopreservation could contribute to DNA damage, lipid peroxidation, and apoptosis, all of this which ultimately decreased the viability and functionality of SSCs [160]. Hence, by improving the protocol of cryopreservation and application of cryoprotectant agents such as antioxidants in cryopreservation, the viability and functionality of SSCs can be restored, thereby resulting in preserving the fertility.

### 6.1. Antioxidants

Although permeable and impermeable cryoprotectant agents have distinct strategies in minimizing the damage during cryopreservation, they are still incapable of eliminating the ROS and preventing the cell from oxidative damage. Therefore, by adding antioxidants to the basic freezing media, it may help in scavenging the ROS and preventing the destructive effects of cryoinjury in cryopreserved cells [161]. Studies have shown that adding an exogenous antioxidant to the freezing extender can improve the quality, functionality, and viability of SSCs after they are thawed. Indirectly, it shows that oxidative stress occurs during cryopreservation, which caused harmful effects to the cells [162,163]. Hence, many studies have investigated the effect of antioxidants in ameliorating the ROS formation to decrease cryoinjury in SSCs [158,162,164].

Previous studies revealed that, after adding antioxidants to the freezing media, the functional rate of proliferation and differentiation of SSCs was increased. These findings are proven by the higher expression of GFRα1 and PLZF in SSCs [158,162,164]. It shows that the antioxidant could restore and improve the functionality of SSCs after the freezing-thawing step in terms of self-renewal and differentiation capabilities. Previous studies found that by adding the catalase (CAT) and α-tricalcium phosphate (α-TCP) to the freezing media, the ROS generation in cryopreserved cells can be controlled, and the detrimental effects of cryopreservation techniques can be lessened [162]. CAT and α-TCP are capable of converting hydrogen peroxide (H_2_O_2_) to water (H_2_O) and oxygen (O_2_), thus eliminating ROS toxicity in the cryopreservation of SSCs [162]. Moreover, Aliakbari et al. [162] reported that utilizing antioxidants in the cryopreservation technique helped to preserve the functionality of SSCs, which was proven by increasing PLZF expression. This reflects that the function of SSCs is normal, as they are capable of self-renewal and differentiation.

Previous researchers also explored the effect of melatonin on reducing the cryoinjury during cryopreservation. Melatonin is the primary hormone secreted by the pineal gland in animals. Melatonin has antioxidant and ROS scavenging capabilities higher than vitamin E does [165,166]. Melatonin protects cells from oxidative stress in numerous ways; the primary mechanism of its action is via its ability in transferring the electrons and hydrogen radical from radical to non-radical molecules [167]. In addition, Rodriguez et al. [168] showed that melatonin controls the gene expression and enzyme activity of the antioxidant enzymes glutathione peroxidase (GSH-Px), superoxide dismutase (SOD), and CAT [123]. Melatonin has therapeutic benefits in clinical medicine and experimental studies, including antioxidant, anti-apoptosis, anti-inflammation, and autophagy effects, as well as modulation of the circadian rhythm [169,170]. Several researchers have demonstrated that melatonin is beneficial in stimulating the self-renewal of SSCs as well as improving the testicular and reproductive system function in men [171,172,173]. In addition, melatonin functions as a cryoprotectant in spermatogonia-containing freezing media and favorably influences male fertility preservation [174,175,176]. Previous studies have demonstrated that the addition of melatonin to the culture media increased the function of SSCs in mice and goats [158,164]. The level of ROS was significantly reduced in the melatonin-treated group. Hence, melatonin played a greater role as an antioxidant in reducing the ROS formation due to cryopreservation [158,177]. Moreover, the integrity function of SSCs was also found to be highly increased in SSCs. This is proven by the increased expression of GFRα1 and PLZF, markers for self-renewal and differentiation [158,164]. Table 1 shows the antioxidant function in the cryopreservation of SSCs and the usage of PLZF and GFRα1 in assessing the functionality of SSCs.

### 6.2. Cryopreservation Protocol Aspect

The development of cryopreservation procedures for SSCs has relied on uncontrolled slow-freezing and fast thawing. These approaches have been demonstrated to be efficient for the long-term storage of SSCs capable of restoring fertility in infertile recipient mice following thawing and transplantation [178,179]. However, a typical side effect of cryopreservation employing uncontrolled slow freezing is the development of cryoinjury due to osmotic stress, which results in a poor survival rate of SSCs. Recent studies also indicated that cryogenic injuries due to the production of ice crystals might unfavorably affect testicular tissue during the freezing phase in cryopreservation, leading to osmotic stress in the sperm plasma membrane [24,180,181,182]. Extreme low and high cooling rates during cryopreservation could minimize the ability of the cells to regulate the intracellular water loss moderately [104]. For instance, the duration for water to leave the intracellular area before vitrification occurs during rapid cooling is insufficient, resulting in a high cellular water content [183]. This eventually leads to the formation of intracellular ice crystals, resulting in cell apoptosis [104,184]. Consequently, improving the cryopreservation protocol, with the use of cryoprotectant agents, the application of chemical tools, and the maintenance of optimal temperature and time equilibrium rate, might increase the proliferation rate of SSCs [185,186,187,188].

Cell encapsulation is currently often employed to facilitate ready-to-use materials for tissue engineering applications or to provide above-zero cell storage for short time periods, which is up to two weeks. Many previous studies have demonstrated that encapsulated cells perform better in cryopreservation than non-encapsulated cells do [189,190,191,192]. However, the cell encapsulation tool is considered new in the cryopreservation of SSCs as compared to other targeted cells such as red blood cells, sperm, and dorsal root ganglion. Pirnia et al. [186] found that there was a normal expression of PLZF in mice SSCs encapsulated in alginate hydrogen after cryopreservation. These findings indicate that freeze-thaw with cells encapsulated with alginate does stimulate the differentiation of SSCs and the self-renewal ability is preserved.

Cryoprotectants chemicals are utilized to prevent the development of ice, which damages biological tissue [193]. Serum acts as a cryoprotectant in freezing media, which is able to enhance the cryopreservation of SSCs. During cryopreservation, serum works as a buffer to control the osmotic stress, preserve the cellular membrane, and limit the risk of crystallization or recrystallization [187,194]. However, serum has limitations such as an undetermined composition, inter-variation among the batch, and the possibility of contamination. Consequently, it could lead to inconsistent results as well as an increase in time and cost. Therefore, serum replacement application is a new targeted approach to improve the successful rate of cryopreservation in SSCs. Jung et al. [187] selected recombinant human serum albumin (rHSA) because it reduced the contamination rate, and it is structurally and functionally similar to plasma-derived HSA. Plasma-derived HSA is the most effective serum replacement for the cryopreservation of hematopoietic stem cells, human umbilical cord-derived mesenchymal stem cells, and bone marrow-derived mesenchymal stem cells [195,196,197]. They found that the recovery rate and proliferation capacity after freeze-thawing were significantly increased in all treatment groups. The expression rates of PLZF and GFRα1 were normal, which indicated no alterations in the self-renewal and differentiation potential of the treatment group. However, more research needs to be carried out to prove the benefit of rHSA, especially in the cryopreservation of SSCs as well as the assessment of cell viability, integrity, and functionality.

In addition, dimethyl sulfoxide (DMSO) and glycerol are the most commonly used cryoprotectants in cryopreservation [198]. These chemicals are added to the cells in reducing osmotic stress due to extreme temperature changes in cryopreservation [199]. Sperm cryopreservation usually involves freezing semen in straws or pellets at −196 °C using liquid nitrogen [182,200]. Storing spermatozoa at higher temperatures has been shown to improve sperm survival once combined with cryoprotectants [201,202,203,204]. A previous study discovered that there was no significant difference in terms of sperm viability of ram spermatozoa stored at 23 °C for 24 h and at 4 °C for 48 h as compared to fresh spermatozoa. Furthermore, a higher fertility rate was indicated for spermatozoa stored at 23 °C and 4 °C when compared to spermatozoa that were cryopreserved at −196 °C for 24 h, with the addition of citrate–glucose–egg-yolk-based cryoprotectant in freezing media [204]. Therefore, the usage of cryoprotectants in sperm cryopreservation in different storage temperature and duration plays an important role in ensuring the quality and integrity of the sperm.

Cryoprotectants are divided into two groups: those that either allow the water to pass through the cells and those that do not. Small molecules such as DMSO and ethylene glycol (EG) can pass through cell membranes and form hydrogen bonds with water molecules. This lowers the freezing point and stops ice crystals from forming inside and outside the cells, which can cause damage during cryopreservation [158]. DMSO is classified as a permeable cryoprotectant agent and is commonly employed for the cryopreservation of murine and human germ cells. DMSO has also been demonstrated to achieve a superior preservation success rate. However, the cellular viability after thawing could be enhanced via the addition of additive cryopreservation agents [179,205,206].

Moraveji et al. [207] demonstrated a freezing protocol that allowed rapid absorption of DMSO inside small fragments of the testis, resulting in reduced cytotoxicity of testicular cells and thus increased the viability of SSCs. Normal expressions of PLZF and GFRα1 were detected in the SSCs showing normal cell function [207]. A previous study also demonstrated that in vitro spermatogenesis utilizing a three-dimensional (3D) culture method resulted in the existence of spermatogonial cells in the testes of chemotherapy-treated prepubertal cancer patient boys (PCPBs) [208]. After 11 months of cryopreservation, isolated testicular cells were cultured in a methylcellulose culture system (MCS)-containing StemPro enhanced with growth factors for 5–15 weeks, and the cells showed normal expression of PLZF and GFRα1. These cells eventually were found to be capable of growing in vitro to various spermatogenic phases, such as the production of sperm-like cells [208].

The optimum equilibration of cryoprotectants has emerged as a viable strategy for enhancing the cryopreservation effectiveness of stallion sperm or mouse embryos [209,210,211]. Not only that, the appropriate equilibration time and temperature may increase the efficacy of cryoprotectants and may be a useful strategy for the cryopreservation of SSCs. Based on the previous study done by Jung et al. [188], about 20 min is required for the SSCs to be in the media enriched with DMSO + trehalose media at the temperature of 4 °C before the freezing step is performed. This step could enhance the survival rate of SSCs. Furthermore, to verify the stemness of SSCs, the researchers cultured the cells for 1 week, and they found an increase in the proliferation rate. The researchers also analyzed the expression of self-renewal and differentiation markers GFRα1 and PLZF using the immunofluorescence method. The results showed that the expression of GFRα1 and PLZF was normal. This indicated that the unique characteristics of spermatogonia itself were preserved [188]. Table 2 shows the cryopreservation techniques of SSCs and the usage of PLZF and GFRα1 for assessing the functionality of SSCs.

## 7. Discussion

The cryopreservation of SSCs is one of the technologies involved in infertility treatment. The development of this technology has given hope to a group of patients (cancer survivors, individuals with KS, and patients with idiopathic non-obstructive azoospermia) who may benefit from regenerative treatments utilizing SSCs technology in vivo or in vitro when they become available in the future [185]. Despite the efficacy of isolation and transplantation of SSCs in various animal species, SSCs therapy has not yet been used to retain fertility in humans [185,213]. However, a systematic review by Zarandi et al. [185] reported two recent studies that attempted to differentiate germ cells in vitro by utilizing tissue from cryopreserved testicles, and the findings showed promising results that might be used in the future for infertility treatment.

SSCs are the most significant cell type in the testis, as they are responsible for spermatogonial pool maintenance and sperm production. Depending on the freezing procedures and types of cryoprotectant, the viability and functionality of preserved SSCs must be evaluated [185]. Various methods have been used to identify the SSCs marker specifically for self-renewal and differentiation such as polymerase chain reaction (PCR), western blot, immunofluorescence, and immunohistochemistry [158,162,186,187,188,207,208]. Findings showed that the detection of markers by using the PCR method provided the most significant results in terms of the expression level for PLZF and GFRα1 [162,186,208,212]. Even though the immunofluorescence staining method is the most common method used by researchers, the expression of the markers PLZF and GFRα1 was recorded to be normal [158,187,188,207,208]. Theoretically, the PCR method is more sensitive and specific because the marker detection is at the gene level that measures the RNA expression [214]. The expression of the gene indicates the possibility of the gene being transcripted and translated to a functional protein. In comparison with immunofluorescent staining methods, the detection level is at the protein level. The PCR method seems to be the better method in terms of the detection of markers of SSCs, and it can give detailed information at the molecular level for monitoring the alteration or changes in SSCs. Other techniques, such as xenografting the tissue and transplanting the cells retrogradely into the seminiferous tubules, were also used by researchers to examine the presence and function of these cells [185]. There are also biomarkers that can be used for assessing the functionality of preserved SSCs such as GFRα1 and PLZF. GFRα1 and PLZF were identified as significant markers where both markers are essential for self-renewal [27,28,29,30,31]; however, only PLZF is involved in initiating the differentiation of spermatogonia [33,34,36,62].

Similar to cryopreserved sperm, the cryopreservation of SSCs results in cryoinjury. It is due to an increase in physiological ROS generation and the limitations of procedures such as freeze-thawing technique, resulting in reducing their ability in self-renewal and differentiation as well as their survivability. Based on our literature search, oxidative stress is only reported to occur in the cryopreservation of spermatozoa. It is proven by a reduction in certain types of antioxidant levels and an increase in lipid peroxidation [215,216,217,218]. Interestingly, no data were reported on the GFRα1 and PLZF expression levels. This might be because the functionality of the sperm are not influenced by both molecular proteins, GFRα1 and PLZF. The spermatozoa are the end-product of spermatogenesis; therefore, self-renewal and differentiation are not necessary for cell integrity and functionality.

The physiological production of ROS during cryopreservation may have contributed to DNA damage, lipid peroxidation, and apoptosis, which lowered the survivability of functional SSCs [160]. Therefore, by utilizing antioxidants in the media for cryopreservation, the functionality and viability of SSCs can be restored. Previous studies showed that antioxidants such as CAT, α-TCP, and melatonin protect the SSCs from cryoinjury, resulting in retaining the cell function, which includes the self-renewal and differentiation abilities proven by significantly increased expression of PLZF and GFRα1 [158,162,164]. Antioxidants can reduce ROS formation by their ability in converting the H_2_O_2_ to H_2_O and O_2_. The reduction in ROS formation in cryopreservation could enhance the functionality and integrity of SSCs.

When SSCs were subjected to cryopreservation procedures, significant fluctuations in temperature, ice crystal formation, and osmotic stress along the cells resulted in increasing the risk of cells surviving and reducing their functionality. Therefore, the usage of cell encapsulation and the equilibration time and temperature influence the integrity, viability, and functionality of SSCs upon cryopreservation [185,186,187,188]. Most of the studies assessed the functionality of SSCs such as proliferation, differentiation, and self-renewal capabilities. Kabiri et al. [219] determined the expression of MAGE-A4 and Oct4 for assessing the viability and proliferation of SSCs from prepubertal boys scheduled for gonadotoxic treatment. However, only a few studies used PLZF and GFRα1 as biomarkers for assessing SSCs function [186,187,188,207,208]. Therefore, we postulated that the PLZF and GFRα1 can be used as specific biomarkers for assessing the self-renewal and differentiation of SSCs during cryopreservation while considering the equilibrium of time and temperature as well as the usage of antioxidants, cryoprotectant agents, and cell encapsulation. More research is required to investigate the importance of PLZF and GFRα1 in the development of a cryopreservation protocol.

Furthermore, PLZF and GFRα1 are known to be the most specific biomarkers to identify the functionality of SSCs. These biomarkers are specific towards type A_single_ spermatogonia (or type A_dark_ spermatogonia), which is responsible for the self-renewal process [36,220,221]. Therefore, by measuring and evaluating these markers after cryopreservation will help in isolating the best cells for further transplantation for the continuation of spermatogenesis. Basically, a testicular tissue biopsy sample was isolated from the patient prior to freezing and storage of the testicular tissues in liquid nitrogen. The usage of antioxidants such as CAT, α-TCP, and melatonin in freezing media will help to mitigate the production of ROS. Besides, rHSA also reduces the osmotic stress that occurs due to crystallization in the cells during freezing. SSCs encapsulation using alginate hydrogel before the freezing phase also helps to minimize the risk in inducing intracellular osmotic stress [186]. Prior to treatment or any transplantation, the SSCs need to be thawed. The SSCs will be cultured, and it is necessary to measure the expression of PLZF and GFRα1 in order to assess the functionality of SSCs. Elevated protein levels of PLZF and GFRα1 indicated that functional and viable SSCs are produced. Then, autologous transplantation of the best selected SSCs into the testis is performed, resulting in the activation of in vivo spermatogenesis in the testis and differentiation into healthy spermatozoa that is suitable for natural fertilization. Figure 2 shows the therapeutic approach for improving the protocol for cryopreservation of SSCs by evaluating the expression of PLZF and GFRα1 as biomarkers.

## 8. Conclusions

Based on our findings, the research is limited regarding the utilization of PLZF and GFRα1 in assessing the functionality of SSCs after cryopreservation. Previous studies have highlighted significant roles of PLZF and GFRα1 as specific biomarkers in identifying the functionality of SSCs prior to cryopreservation. Hence, these two markers have growing potential in future research for evaluating the functionality of SSCs as well as producing high-quality SSCs, thereby achieving a high fertility rate. More clinical cryopreservation experiments involving human SSCs based on these biomarkers need to be performed as the present findings are scarce. In addition, the data documented in this review will be helpful in giving insights to ensure the best optimization of better cryopreservation protocols in assisting human fertility.

## Figures and Tables

**Figure 1 ijms-24-01945-f001:**
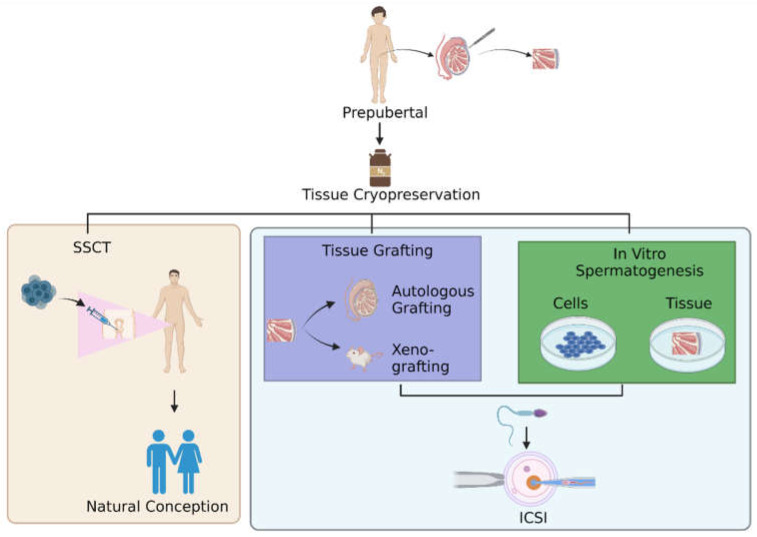
Pathways for restoring fertility. After the thawing step in cryopreservation of testicular tissue, there are three options for men to restore fertility: undergoing SSCT, testicular tissue grafting, or in vitro spermatogenesis. Abbreviation: SSCT; spermatogonial stem cells transplantation, ICSI; intracytoplasmic sperm injection.

**Figure 2 ijms-24-01945-f002:**
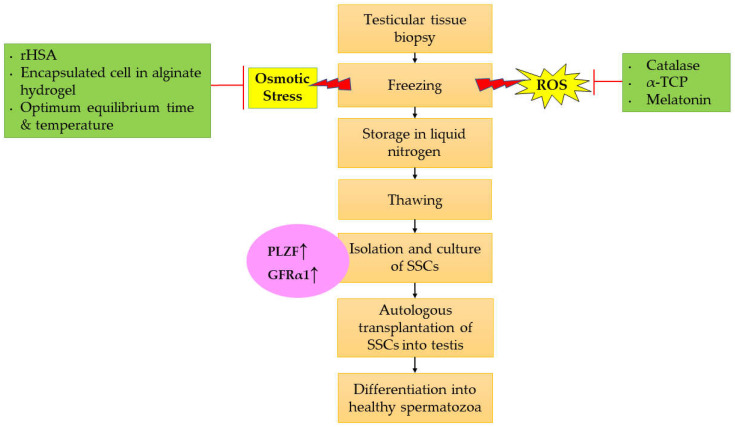
Therapeutic approach for improving the cryopreservation of SSCs via evaluating the expression of PLZF and GFRα1 as biomarkers. Abbreviation: ↑ Increase/enhance.

**Table 1 ijms-24-01945-t001:** Antioxidant function in cryopreservation of SSCs and expression of PLZF and GFRα1 in assessing the functionality of SSCs.

Treatment	Animal/Cell Line	Period of Cryopreserved	Target Markers	Finding	Author
Catalase and α-TCP	Isolated from 3 to 6-day-old male mice	1 week	PLZF	↑ PLZF (*p* < 0.001)	[162]
Melatonin	Isolated from 3–6-day-old male NMRI mice	2 weeks	GFRα1 PLZF	↑ PLZF (*p* < 0.001)	[158]
				↑ GFRα1 (*p* < 0.001)	
				↓ ROS (*p* < 0.001)	
Melatonin	Isolated from 3–6-day-old male BALB-c mice	1 month	PLZF	↑ PLZF (*p* < 0.05)	[164]

Abbreviations: ↑: Increase; ↓: Decrease, α-TCP: α-tricalcium phosphate.

**Table 2 ijms-24-01945-t002:** The difference in cryopreservation protocols for assessing the functionality of SSCs.

Treatment	Animal/Cell Line	Period of Cryopreserved	Target Markers	Finding	Author
SSCs encapsulated in alginate hydrogel	Isolated from 6-day-old male Balb/C mice	-	PLZF	N: PLZF	[186]
5% Recombinant human serum albumin (rHSA)	Isolated from 6–8 day-old male C57-GFP mice	1 month	PLZF GFRα1	↑ Recovery rate (*p* < 0.001)	[187]
				↑ Proliferation capacity (*p* < 0.001)	
				N: PLZF and GFRα1	
Equilibrium time (20 min) and equilibrium temperature (4 °C and RT)	Isolated from 6- to 8-day-old C57-GFP mice	1 month	PLZF GFRα1	↑ Survival rate in combination of 10% DMSO and 200 mM trehalose (*p* < 0.05)	[188]
				↑ proliferation rate after culture for 1 week at 20 min of equilibration at 4 °C (*p* < 0.05)	
				N: PLZF and GFRα1	
2 years of long-term-cultured SSC line	Isolated from 6-month-old fry fish (*Opsariichthys bidens*)	2 weeks	GFRα1	Differentiation of ObSSCs into more mature sperm with slight wobbling characteristics in their tails after 16 days of coculture	[212]
				↑ GFRα1	
Isolation protocol 2 (IP2) and freezing protocol 3 (FP3) with DMSO	Isolated SSCs from testicular samples from brain-dead men	-	PLZF GFRα1	N: PLZF and GFRα1 ↑ SSCs viability and colony formation	[207]
Controlled slow freezing (CSF) with 5% DMSO, 10% HSA	Testicular tissues obtained from male prepubertal cancer patients (age range 6–13 years)	11 months	PLZF GFRα1	Sperm-like cells were successfully developed	[208]
				N: PLZF and GFRα1	

Abbreviations: N: Normal; ↑: Increase; DMSO: dimethyl sulfoxide; rHSA: recombinant human serum albumin; GFP: green fluorescence protein; RT: room temperature; ObSSC: *Opsariichthys bidens* spermatogonia stem cell; IP2: isolation protocol 2; FP3: freezing protocol 3; CSF: controlled slow freezing.

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
