# Peer review of "The Role of Promyelocytic Leukemia Zinc Finger (PLZF) and Glial-Derived Neurotrophic Factor Family Receptor Alpha 1 (GFRα1) in the Cryopreservation of Spermatogonia Stem Cells"

_ijms, 2023, doi:10.3390/ijms24031945_

Round 1

Reviewer 1 Report

Questions and Recommendations

The presented review is focused on the role of PLZF and GFRα1 biomarkers as indicators of self-renewal and differentiation ability of cryopreserved SSCs.  The authors presented that these two biomarkers might indicate the functionality of the stem cells after cryopreservation.

The review is very interesting and well written. It summarizes significant biological findings about the presented biomarkers, which might be assessed in order to evaluate the usefulness of cryopreservation protocols.

Below are listed minor comments that should be corrected and clarified before the final acceptance of paper:

Abstract:

Lines 23-24: Please correct the sentence as follows: “The therapeutics intervention such as use of antioxidants and enhancement in cryopreservation protocols such as cell encapsulation, cryoprotectant agents and…”

1. Introduction:

Line 54: Please add space “to 29”.

Line 76: There should be a comma: “[27,28]”.

Line 95: Did the authors mean: “for isolation of potential SSCs”?

2. Spermatogenesis:

Line 126: Please correct: “testosterone is essential for spermatogenesis, which occurs”.

Lines 127-128: Did the authors mean: “this hormone is also essential for the SSCs to be functional after cryopreservation”?

Line 129: Please explain the abbreviation “FSHR” within the text.

Line 185: Please add sign “-“ to phenotype (GFRα1+/PLZF+/cKIT-).

3. Spermatogonia Stem Cell’s Biomarkers:

Lines 217-218: Please correct sentence as follows: “(GFRα1) plays a role for the growth factor GDNF (glial cell line-derived neurotrophic factor) as a co-receptor”.

Line 2019: Please correct “transection” to “transfection”.

Lines 267-268: Please add missing term, maybe “factor?”: “PLZF known as key cell-autonomous factor in promoting the ability…”

Line 273: Please remove “1)” after term “(GNDF).

Line 284: Please correct as follows: “PLFZ is able”.

Lines 289-290: Please rewrite the following sentence to make it clear: “during the differentiation of SSCs, it involves the androgen level and Sertoli cells.”

Lines 299-230: Please correct grammar: “suggest that by blocking the androgen” to “suggest that blocking of the androgen”.

3. Cryopreservation of SSCs:

Line 307: This should be chapter number 4, as chapter 3 is titled: “Spermatogonia Stem Cell’s Biomarkers”.

Line 356-357: Please the rewrite the following sentence to make it clear: “This shows the continual long-term growth in the number of SSCs in culture now permits the introduction of genetic alterations through the germ line which suits clinical experimental needs in producing somatic cells.”

4. Therapeutic Approach Improving the SSCs cryopreservation:

Line 366: This should be chapter 5. Please correct the numbering of chapters and subsections (Lines 384 and 444).

Lines 367-368: Please correct “elicit” to elicits” and “sperm cryopreserved” to “sperm cryopreservation” or “cryopreserved sperm”.

Line 368: Please rewrite as follows: “Despite the damage caused by the process of freezing and thawing itself, ...”

Line 377: Please correct “increased” to “increase”.

Line 378:  Please remove “of” in front of “ROS production”.

Line 380: Please correct “cryopreserved” to “cryopreservation”.

Line 384: Please remove dot from the title of subsection.

Lines 405-408: Please rewrite the following sentence to make it clear: “Moreover, Aliakbari et al. [113] also reported that by utilizing the antioxidants in cryopreservation technique preserved the functionality of SSCs proven by increasing of PLZF expression; reflecting the function of SSCs are normal which is capable in self-renewal and differentiation.”

Line 416: Please remove terms “found that” from the sentence.

Line 417-418: Please correct sentence as follows: “The self-renewal capacity of spermatogonia was also reported to be improved by the increased expression of GFRα1.”

Line 441: Please correct “increase” to “increased”.

Line 464 (Table 1): N: Normal is not mentioned within the data in table.

Lines 499-500: Please rewrite the following sentence to make it clear: “Storing spermatozoa at higher temperatures is capable in preserving for the better sperm survival with the additional of cryoprotectants.”

Lines 504-505: Please correct “additional” to “addition”.

Line 525: Please correct “significantly increased” to “significant increase”.

Lines 530-532: Please rewrite the following sentence to make it clear: “The researchers found out that the progeny of the recipient expressed GFP-positive spermatogonia cells in 200 Mm trehalose for 3 months of the freezing period.”

Line 540: Please explain the abbreviation “SPGCs” within the text.

Line 568: Please correct “taken placed” to “is performed”.

Discussion:

Line 580: This should be chapter 6.

Line 581: Please correct “involves” to “involved”.

Lines 606-608: Please rewrite the following sentence to make it clear: “Based on our literature search, the oxidative stress only being reported to occur in spermatozoa cryopreservation proven by reduction in certain type of antioxidants levels and increased in the lipid peroxidation.”

Line 627: Please correct “encapsulated” to “encapsulation”.

Line 661 (Figure 1): Please provide figure with better resolution.

Line 664: This should be chapter 7.

References:

References 6, 7 and 179: The author whole names are in upper case. This should be corrected.

Author Response

Thank you very much.

Reviewer 2 Report

Iwant to thank the authors for the great efforts conducted to produce this review, I have some recommendations. 

1- in the introduction, please modify the definition of the secondary infertility 

2- in the introduction, when authors start to define the type of sperm abnormalities l44-l46, it's better to add also the two terms teratozoospermia and asthenozoospermia.

3- the passage between animal models and human is not well presented.

4- I suggest adding a paragraph concerning the clinical cryopreservation experiments involving human and animal SSCs. 

5- I suggest adding a paragraph concerning the methods of measuring and evaluating PLZF and GFRα1. 

6- clinical data regarding the use of SSCs and its outcome is also requested.

Author Response

Thank you very much.
